# Citrinin Exposure in Germany: Urine Biomarker Analysis in Children and Adults

**DOI:** 10.3390/toxins15010026

**Published:** 2022-12-30

**Authors:** Gisela H. Degen, Jörg Reinders, Martin Kraft, Wolfgang Völkel, Felicia Gerull, Rafael Burghardt, Silvia Sievering, Jennifer Engelmann, Yvonni Chovolou, Jan G. Hengstler, Hermann Fromme

**Affiliations:** 1Leibniz Research Centre for Working Environment and Human Factors (IfADo), Ardeystrasse 67, D-44139 Dortmund, Germany; reinders@ifado.de (J.R.); hengstler@ifado.de (J.G.H.); 2State Agency for Nature, Environment and Consumer Protection North-Rhine Westphalia, Department of Environmental Medicine, Wallneyer Straße 6, D-45133 Essen, Germany; martin.kraft@lanuv.nrw.de (M.K.); silvia.sievering@lanuv.nrw.de (S.S.); info@lanuv.nrw.de (J.E.); yvonni.chovolou@lanuv.nrw.de (Y.C.); 3Bavarian Health and Food Safety Authority, Department of Chemical Safety, Toxicology and Exposure Monitoring, Pfarrstraße 3, D-80538 München, Germany; wolfgang.voelkel@lgl.bayern.de; 4Landeslabor Berlin-Brandenburg, Fachbereich IV-4, Umweltbezogener Gesundheitsschutz, Rudower Chaussee 39, D-12489 Berlin, Germany; felicia.gerull@landeslabor-bbb.de (F.G.); info@landeslabor-bbb.de (R.B.); 5Institut und Poliklinik für Arbeits-, Sozial- und Umweltmedizin, Klinikum der Ludwig-Maximilians-Universität München, Ziemssenstraße 1, D-80336 München, Germany; hermann.fromme@med.uni-muenchen.de

**Keywords:** biomonitoring, citrinin, dihydrocitrinone, exposure, mycotoxin, nephrotoxin

## Abstract

Citrinin (CIT), a mycotoxin known to exert nephrotoxicity, is a contaminant in food and feed. Since CIT contamination is not regularly analyzed, data on its occurrence and especially levels in food commodities are insufficient for conducting a conventional exposure assessment. Yet, human biomonitoring, i.e., an analysis of CIT and its metabolite dihydrocitrinone (DH-CIT) in urine samples allows to estimate exposure. This study investigated CIT exposure in young (2–14 years) and adult (24–61 years) residents of three federal states in Germany. A total of 179 urine samples from children and 142 from adults were collected and analyzed by a targeted LC-MS/MS based method for presence of CIT and DH-CIT. At least one of the biomarkers was detected and quantified in all urines, which indicated a widespread dietary exposure to the mycotoxin in Germany. Interestingly, the biomarker concentrations of CIT_total_ (sum of CIT and DH-CIT) were higher in children’s urine (range 0.05–7.62 ng/mL; median of 0.54 ng/mL) than in urines from adults (range 0.04–3.5 ng/mL; median 0.3 ng/mL). The biomarker levels (CIT_total_) of individual urines served to calculate the probable daily CIT intake, for comparison to a value of 0.2 µg/kg bw/day defined as ‘level of no concern for nephrotoxicity’ by the European Food Safety Authority. The median exposure of German adults was 0.013 µg/kg b.w., with only one urine donor exceeding this provisional tolerable daily intake (pTDI) for CIT. The median exposure of children was 0.05 µg/kg bw per day (i.e., 25% of the pTDI); however, CIT exposure in 12 individuals (6.3% of our study group) exceeded the limit value, with a maximum intake of 0.46 µg/kg b.w. per day. In conclusion, these results show evidence for non-negligible exposure to CIT in some individuals in Germany, mainly in children. Therefore, further biomonitoring studies and investigations aimed to identify the major sources of CIT exposure in food commodities are required.

## 1. Introduction

Citrinin (CIT) is a mycotoxin produced by several species of the genera *Penicillium* and *Aspergillus*, found in various climate zones [1,2]. CIT is a known contaminant in various grains and cereal-based products, and often along with ochratoxin A (OTA), another more potent nephrotoxic mycotoxin [3,4,5]. Rice fermented with *Monascus* spp., so called red yeast rice, used in Asia for food coloring and also marketed in Europe as cholesterol lowering food supplement, can contain very high levels (>2000 µg/kg) of CIT [6,7,8,9]. In 2019 the maximum level for CIT in red yeast rice based food supplements has been reduced in Europe to 100 µg/kg [10]. However, so far there is no regulation on maximal CIT levels in cereals and other food commodities to protect consumers against an undesirable dietary intake.

The CONTAM-Panel of the European Food Safety Authority (EFSA) evaluated risks related to the presence of CIT in food and feed in 2012, and noted research needs regarding dietary exposure and some uncertainty on potential carcinogenicity and genotoxicity; yet the Panel could derive a ’level of no concern for nephrotoxicity’ of 0.2 µg/kg body weight/day as a provisional tolerable daily intake (pTDI) value for humans [4]. Since then, due to improved analytical methods for CIT detection in various matrices, more data on its occurrence and levels in food and feed have been generated [2,7,8,11,12]. Yet, data on CIT presence in major food commodities is still scarce, and this hampers a conventional exposure assessment, which combines such contamination data with food consumption information in various groups of the population.

Human biomonitoring is widely applied to investigate mycotoxin intake from all sources and routes of human exposure by analysis of biomarker concentrations in biological fluids [13,14]. The analysis of CIT and its metabolite dihydrocitrinone (DH-CIT), mainly in urine as matrix of choice due to ease of collection, is a valuable approach for investigating dietary exposure from all sources. First reports are those by Blaszkewicz et al. [15] and by Ali et al. [16] on biomarker occurrence in urines from German adults, and a comparative study of the urinary biomarker excretion patterns in Bangladesh, Germany, and Haiti by a multi-mycotoxin method [17]. More recent results of CIT biomarker analysis in cohorts from several countries, by targeted or by multi-mycotoxin methods, have been reviewed [18,19,20]; these data show widespread exposure to this nephrotoxic food contaminant, as well as variations in the occurrence and urine levels of CIT and DH-CIT in different parts of the world.

Such urine biomarker data can be also used for calculating the mycotoxin’s probable daily intake (PDI) since information on kinetics and urinary excretion rates in humans are now available, which then allows to assess risks by comparing the estimated PDIs for CIT to the ’level of no concern for nephrotoxicity’ as provisional tolerable daily intake (pTDI) value [21]. This is of considerable interest, also in the light of co-occurrence with the nephrotoxic mycotoxin OTA in foods and in human fluids [4,11,22,23,24]. However, biomonitoring data on CIT are limited, and only a few studies so far include children cohorts [20,23,25]. The present study is the first one on CIT biomarkers in urines from German children and adults collected in three federal states. The results of our survey states are discussed in the context of other data on CIT biomarker levels and biomarker-based intake estimates and in relation to the provisional TDI value for CIT.

## 2. Results

Urine samples collected in three federal states of Germany (Bavaria, Berlin and North-Rhine Westphalia) from children (2–14 years) and adults (24–61 years). The majority of samples were spot urine samples, but 10 children provided both morning urines and whole day urines. Biomarker analysis used IAC for enrichment of analytes and LC-MS/MS analysis with isotope labeled internal standards for CIT and DH-CIT (see Section 5.3). In all urine samples, at least one of the biomarkers could be found: CIT was present in a range of <LOD to 1.43 ng/mL (mean 0.04 ± 0.1 ng/mL) and DH-CIT in a range of 0.04–7.44 ng/mL (mean 0.64 ± 0.78 ng/mL). As seen in other biomonitoring studies, urine levels of DH-CIT were often higher than those of CIT. Since the sum of parent mycotoxin and its metabolite (C_total_) in urine best reflects exposure to CIT, the results for our study group at large and for subgroups are presented in this way: the levels of C_total_ expressed as ng/mL urine are summarized in Table 1.

The biomarker levels determined in all urine samples indicate variable, but widespread dietary CIT exposure of our entire study group. Considering different age groups, we noted that the average (mean and median) CIT_total_ concentrations in urines of children were higher than in those of adults. This is readily apparent for urines of children and their adult family members in Bavaria and Berlin; in North Rhine-Westphalia (NRW) only urines of children were available (Table 1). For an easier comparison between age groups the urine biomarker values are depicted in a box plot (Figure 1).

The ranges of biomarker concentrations show some overlap for adults and children, yet the median value in the young is significantly higher than in adult urine donors.

The NRW group consisted of kindergarden children (*n* = 50) who provided morning urines on one day, and 10 individuals where 24 h-urines were also available. This allowed some insights how biomarker levels vary between individuals and sampling method. Urine CIT_total_ concentrations in this subgroup are depicted in Figure 2.

For the majority (6/10) of children, morning urines were found to contain higher levels of CIT_total_ than 24 h urines; in three children 24 h urines showed slightly higher biomarker levels, differing only by a small factor (<2) from the spot urine sample. Hence, for the majority of this small subgroup the results of morning spot urine analysis may not underestimate mycotoxin exposure. The view that biomonitoring reflects variable dietary CIT intake is supported by CIT biomarker analysis data for two adults who collected repeatedly spot urine samples during a period of 7 days and 7 weeks [18].

For 10 children who provided 24 h urines, individual biomarker concentration, total urine volume, and body weight served to calculate their probable daily intake (PDI) of CIT, assuming a median excretion rate of 40.2% (see Section 5.4). The PDIs in this subgroup ranged between 0.042 to 0.166 µg/kg b.w. which equals 20.5 to 82.8% of the provisional daily intake (pTDI) value of 0.2 µg/kg b.w. set by EFSA for CIT as ’level of no concern for nephrotoxicity’.

The same approach was then used for all biomarker data obtained from spot urine samples, yet applying age-adjusted daily urine volumes in the calculation of PDI values (see Section 5.4 for details). The estimated CIT exposure in our cohort is summarized in Table 2.

The median CIT exposure in German adults of 0.013 µg/kg b.w. equals ≤ 7% of the provisional tolerable daily intake (pTDI), a value only surpassed in one individual. In children, the median exposure of 0.05 µg/kg b.w. equals 25% of the pTDI, with 12 individuals at/above this value. The highest CIT intake in children equals 231% of the ’level of no concern for nephrotoxicity’ of 0.2 µg/kg b.w.; an exceedance of this value is considered an undesirable mycotoxin exposure level. The CIT exposures of German adults and children from our survey is depicted in Figure 3 to better illustrate the distribution of daily intake values.

## 3. Discussion

The knowledge on dietary CIT exposure in humans is rather limited as this contaminant is not regularly analyzed in food or feed, a requirement only in place for regulated mycotoxins. Legal limits for CIT are set for food supplements based on rice fermented with red yeast *Monascus purpureus* [10], a potential source of CIT exposure for some adults in Europe [9]. Yet, the general population in Europe consumes foods which may contain this mycotoxin: CIT can be found in grains (e.g., maize, oats, rice, wheat) and other plant products (fruits, herbs, olives, spices and nuts), showing a wide distribution across different geographical areas of the world and in concentrations ranging from a few µg/kg up to 5000 µg/kg depending on the commodity (data reviewed by [2,4,8]). A survey in eight Europe countries detected CIT in a low percentage of foods, with maximum concentrations of 155 µg/kg and 5.7 µg/kg in cereals and cereal-based samples from retail [7]. On the other hand, for maize harvested in Serbia between 2012 and 2015, occurrence rates and mean CIT concentrations differed significantly between production years, being highest (950 ± 2872 μg/kg) in samples from the 2015 maize growing season [26]. Such seasonal fluctuations in the contamination of crops with mycotoxin producing fungi are also well known for other (regulated) mycotoxins [27,28]. This creates uncertainty in assessing human dietary mycotoxin exposure, including CIT where regular surveillance of this contaminant in food commodities is lacking.

Biomonitoring studies from Europe, Asia and Africa report the presence of CIT and its metabolite DH-CIT in many urine samples and indicate variable, yet widespread exposure in several countries. High urine biomarker concentrations were measured for adults in Nigeria (mean ± SD for CIT: 5.96 ± 27.43 ng/mL and DH-CIT 2.39 ± 3.56 ng/mL; [29]), and intermediate levels in infants in Zimbabwe (median CIT 1.4 ng/mL and DH-CIT 0.86 ng/mL; [30]). Biomarker levels of adults in Bangladesh varied between seasons (mean ± SD for CIT: 0.59 ± 0.98 ng/mL and DH-CIT 3.18 ± 8.49 ng/mL in winter or mean ± SD for CIT: 0.10 ± 0.17 ng/mL and DH-CIT 0.42 ± 0.98 ng/mL in summer; [31]). Lower biomarker levels were found in Belgium, with mean concentrations of 0.06 ng/mL CIT and 0.75 ng/mL DH-CIT in urines from adults, and 0.03 ng/mL CIT and 0.55 ng/mL DH-CIT in those from children [25]. Also in urines collected earlier from German adults CIT and DH-CIT were present at mean concentrations of 0.03 ± 0.02 ng/mL and 0.10 ± 0.10 ng/mL, respectively [16]. Such differences in biomarker concentrations between population groups as well as seasonal variations observed in Bangladesh (data reviewed in [18,19]) are likely to reflect different levels of CIT contamination in the foods consumed and/or food preferences of the urine donors in various countries.

The results of the present study provide clear evidence for dietary CIT exposure in German adults and in children, with urine biomarker levels indicating clearly higher exposures in children than in adults (Table 1 and Figure 1). The mean and median concentrations of CIT_total_ in these German urines are higher than levels reported for CIT and DH-CIT in Belgium samples and in an earlier study of German adults (see above; [16,25]). Of note, the latter urines were collected in 2013, those in Belgium in 2013 and 2014, whilst the present study analyzed urines collected in 2015 and early 2016. As prevalence and concentrations of mycotoxins in grains can fluctuate considerably from one year to another [26,27,28], a variable contamination in food commodities can be expected and then also in biomarker results. Thus, a biomonitoring study in a population informs about the exposure situation in a given setting, and follow-up analysis is recommended, in particular when data may raise concerns.

To further assess CIT exposure the probable daily intake was calculated for German adults and children (Table 2), based on individual biomarker levels, individual body weights, age-adjusted daily urine volumes and a median daily CIT excretion rate of 40.2% [21]. For 10 children, the concentrations of CIT_total_ (sum of CIT and DH-CIT) in urine (Figure 2), the total volume of 24 h-urines and their individual body weights were used in the calculation. The PDIs in this subgroup range from 0.042 to 0.166 µg/kg b.w. which represents 20.9 to 82.9% of the pTDI value for CIT defined as ’level of no concern for nephrotoxicity’ [4]. In the data set for all children, the median CIT intake estimate of 0.05 µg/kg b.w. equals 25% of the pTDI, but 12 individuals have CIT intakes at and above this level, with the highest exposure equal to 231% of the pTDI (Table 2). The data set for adults shows clearly lower intake estimates, with a median CIT exposure at 6.5% of the pTDI, and only one individual exceeding this value (Figure 2). Our results resemble findings in a recent study in Italy [20] where average CIT exposure of children (*n* = 20, <18 years) is also higher than that of adults (*n* = 170, 18–65 years), and maximal intake estimates surpass the pTDI, indicative of non-negligible CIT exposure in four children.

As pointed out before, intake estimates for mycotoxins on the basis of urine biomarker concentrations involve some degree of uncertainty: usually spot urines or first morning voids are analyzed rather than 24 h-urines; additionally, absorption, metabolism, and the rate of excretion (i.e., % of ingested mycotoxin excreted as parent compound or metabolites) can vary in individuals [19,20,21]. Yet, the measurement of biomarkers of exposure is the only approach that integrates exposure from all sources and reflects the biologically relevant internal dose. Available biomarker-based estimates of CIT intake provide a reasonably good approach to conclude on prevalence and degree of exposure in a group or population, but give no clues on the foods which contribute.

At present, we may speculate about the dietary sources in our group of German adults and children based on recent studies in Belgium and the Netherlands, as CIT food analysis data are lacking so far for Germany. A total of 357 samples belonging to different food groups, collected in Belgium supermarkets between march 2017 and august 2019, were analyzed for CIT and OTA occurrence [11]. CIT was found in a large number of cereal-based products at mean concentrations of 0.73 µg/kg, and a remarkably high level of 22.9 µg/kg in one sample of whole-grain rice. Other food groups with a fairly high prevalence of positive detects were herbs and spices, meat products and meat imitates, nuts and seeds or fruit and vegetable juices, with mean concentrations between 0.14 to 1.44 µg/kg (see Table 1 in [11]). Food analysis data were then combined with food consumption data for deterministic and probabilistic exposure assessments in different age groups: this showed not only a frequent exposure to CIT in the Belgium population, but the estimated intake can also reach levels of some concern [11]. A recent Total Diet study analyzed several mycotoxins, including CIT, in foods and beverages consumed by 1- and 2 years old infants in the Netherlands; exposure was calculated by combining concentration ranges determined in composite samples with consumption data for various food groups [32]. Whilst the CIT exposure calculations for Dutch infants remained below the pTDI, it is of interest to take note of the food groups which contributed most to the overall intake, namely bread, biscuits, breakfast cereals, chicken, fish and shellfish. However, as yet there is no information on CIT occurrence in foods in Germany. Overall, the biomarker data presented here indicate widespread exposure to CIT in Germany, and at levels that should trigger further efforts to monitor this mycotoxin by complementary approaches, i.e., food analysis and biomarker studies.

## 4. Conclusions

The results of this study show evidence for non-negligible exposure to the mycotoxin CIT in some individuals in Germany, mainly in children. Therefore, follow-up biomonitoring studies and investigations aimed to identify the major sources of CIT exposure in food commodities are required. 

## 5. Materials and Methods

### 5.1. Chemicals and Reagents

CIT (CAS 518-75-2; purity > 98%) was purchased from Sigma-Aldrich (Taufkirchen, Germany) and DH-CIT (CAS 65718-85-6; purity 98.9%) was from AnalytiCon Discovery GmbH (Potsdam, Germany). Stable isotopically labeled standards ((±)-[^13^C_3_]-CIT and (±)-[^13^C_3_]-DH-CIT), synthesized by Bergmann et al. [33] for use in biomarker analysis, were kindly provided by Dr. Benedikt Cramer (Institute of Food Chemistry, University of Münster, Germany). Immunoaffinity columns (IAC) CitriTest^®^ (Vicam^®^, purchased from Ruttmann, Hamburg, Germany) were used for clean-up and enrichment of CIT and its metabolite, as the antibody of this IAC efficiently cross-reacts with DH-CIT [15]. All solvents used to prepare solutions or used as mobile phases in LC-MS/MS analysis were HPLC and LC-MS grade and obtained from Merck (Darmstadt, Germany).

### 5.2. Study Groups and Urines

Urine samples were collected in 2015 and 2016 within an earlier survey aimed to assess the exposure of children and adults to indoor air pollutants in three federal states in Germany. The ethics committee of the Bavarian State Medical Association (Munich, Germany) confirmed the ethical safety (application dated 15 June 2011, ethics committee no. 11053). The responsible data protection officer approved the study in the form it was carried out. All participants signed a written informed content to participate in the study. Some demographic information on urine donors is compiled in Table 3.

Morning spot urines of young children, their siblings and adult family members had been kept for a few hours at +4 °C before storage at −20 °C. The coded urine samples were shipped on dry ice to *IfADo* for biomarker analysis. 

### 5.3. Biomarker Analysis

Aliquots of coded urine samples were analyzed by a validated method which applied immunoaffinity columns (IAC) for clean-up and enrichment of analytes prior to LC-MS/MS analysis [16]. The biomarker analysis was accomplished as published previously [21], with isotope labelled internal standards ([^13^C_3_]-CIT and DH-CIT [^13^C_3_]-DH-CIT), both at a final concentration of 0.5 ng/mL urine. Thus, additional transitions for ^13^C_3_-DH-CIT were measured by a triple quad mass spectrometer (QTrap 5500 from ABSciex, Darmstadt, Germany, equipped with a Turbo V™ Ion Spray source) using the following transitions (268.0 → 178.1 and 268.0 → 224.1 with collision energy of −38 eV for both transitions). Data analysis was done with Analyst software 1.6.1 from AB Sciex (Darmstadt, Germany). Biomarker quantification in sample extracts was based on internal standards, accounting for possible loss of analyte during sample preparation and correcting for matrix effects. Analysis of CIT and DH-CIT in spiked blank urine yielded an LOD of 0.01 ng/mL and an LOQ of 0.03 ng/mL for both analytes. In all urine samples at least one biomarker was found at measurable levels. As reported in the Results, CIT was present in a range of <LOD to 1.43 ng/mL (mean 0.04 ± 0.1 ng/mL) and DH-CIT in a range of 0.04–7.44 ng/mL (mean 0.64 ± 0.78 ng/mL).

### 5.4. Estimate of CIT Intake

CIT exposure in the different study groups was calculated based on the results for individual urine biomarker levels, i.e., the sum of CIT plus DH-CIT concentration (‘total CIT’), an average daily urinary ‘total’ excretion (% of ingested dose), and some additional parameters, according to the following equation for a probable daily intake (PDI):
$$\mathrm{PDI}\;{\left(µg/\mathrm{kg}\;\mathrm{body}\;\mathrm{weight}/\mathrm{day}\right)}^{}=\frac{C\times V\times 100}{W\times E}$$

where C is the urinary total CIT biomarker concentration, V is the average volume of urine excreted in 24 h of 1.43 L for adults [34] and for children age-adjusted values between 0.65 or 1.16 L [35]; W is the individual body weight recorded for adult or children urine donors, and E is the daily urinary mycotoxin excretion rate of 40.2% (the median fraction of an oral CIT dose excreted within 24 h; [21]).

### 5.5. Statistical Analysis

As data were not normally distributed, Mann-Whitney U tests were undertaken to determine the differences in biomarker levels and exposure between children and adults. Calculations were carried out with the GraphPad Version 9.4.1 and significance was assumed for a *p*-value < 0.05.

## Figures and Tables

**Figure 1 toxins-15-00026-f001:**
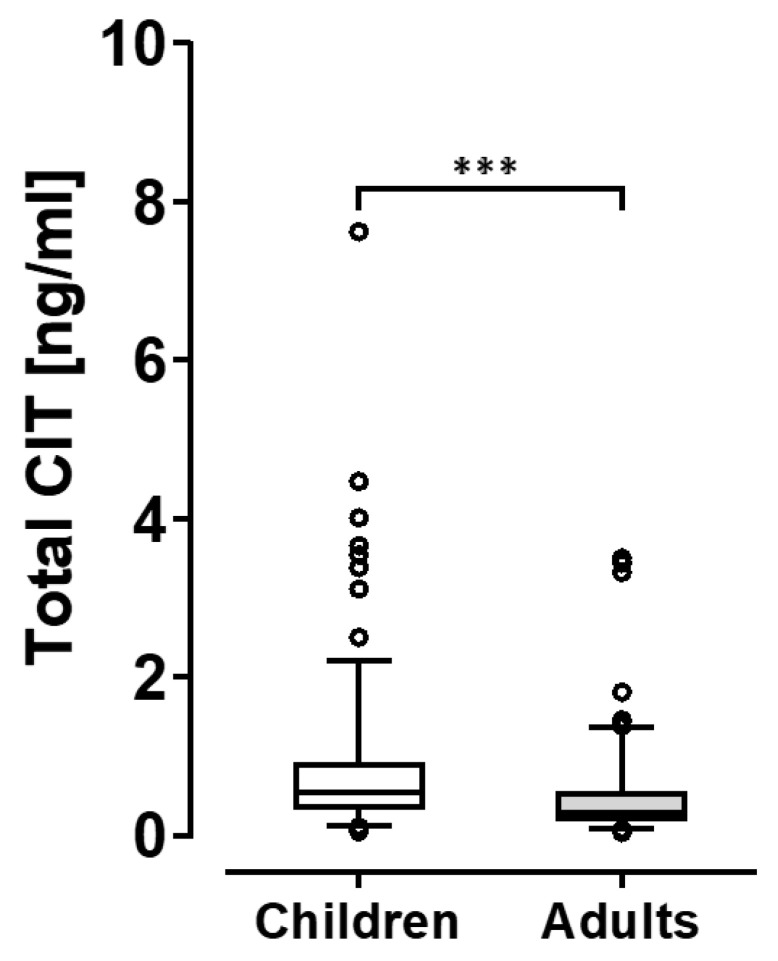
Urine biomarker levels in children of 3 regions and adults in 2 regions. *** denotes a statistically significant difference at *p* < 0.001.

**Figure 2 toxins-15-00026-f002:**
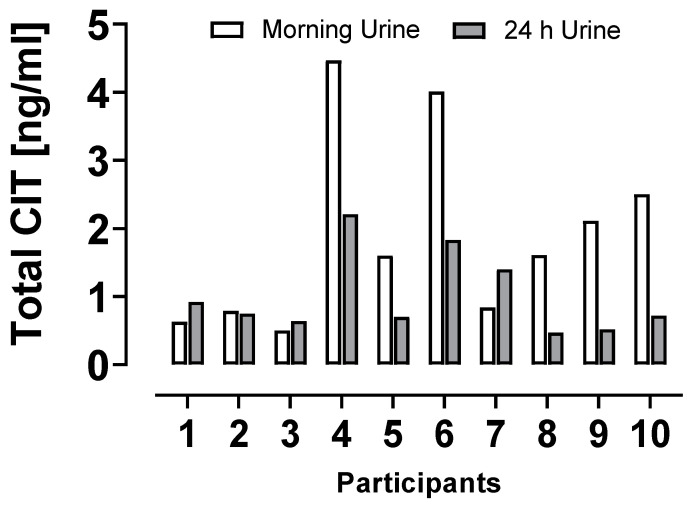
CIT biomarker levels (ng/mL) in morning and 24 h urines from 10 children.

**Figure 3 toxins-15-00026-f003:**
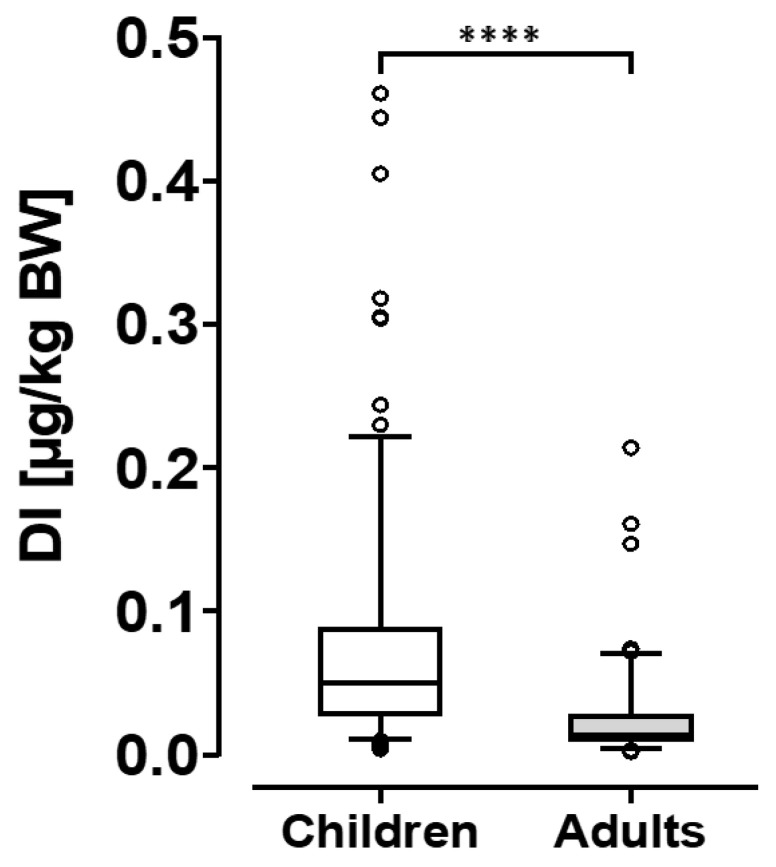
Citrinin exposure (DI = daily intake) in children and adults in Germany. **** denotes a statistically significant difference at *p* < 0.0001.

**Table 1 toxins-15-00026-t001:** Biomarker concentrations (CIT_total_ ng/mL urine) in German children * and adults.

Study Group (*n* = Urine Samples)	CIT_total_ Min–Max	Mean ± SD	Median	P95
Entire study group	0.04–7.62	0.68 ± 0.79	0.43	2.11
Adults (*n* = 142)	0.05–3.50	0.48 ± 0.56	0.30	1.28
Children (*n* = 179)	0.04–7.62	0.83 ± 0.91	0.54	2.20
Bavaria all	0.05–3.66	0.51 ± 0.52	0.35	1.46
Adults (*n* = 76)	0.05–1.81	0.42 ± 0.37	0.30	1.19
Children (*n* = 93)	0.05–3.66	0.60 ± 0.62	0.43	1.66
Berlin all	0.04–7.62	0.74 ± 1.03	0.41	2.54
Adults (*n* = 66)	0.04–3.50	0.55 ± 0.71	0.29	1.44
Children (*n* = 27)	0.13–7.62	1.19 ± 1.49	0.65	2.82
North-Rhine Westphalia (All)	0.23–4.47	1.04 ± 0.89	0.75	2.59
Children (morning urines *n* = 50)	0.23–4.47	1.05 ± 0.94	0.75	3.03
Children (24 h urines *n* = 10)	0.47–2.21	1.02 ± 0.60	0.74	2.04

* The age range of children in the three regions differs considerably (see Table 3).

**Table 2 toxins-15-00026-t002:** Citrinin exposure assessment based on biomarker results as range of probable daily intakes (PDI) and expressed as percentage of the provisional tolerable daily intake (pTDI).

Study Group	Probable Daily Intakes (ng per kg Body Weight)			Percentage of the pTDI (i.e., 200 ng/kg bw *)		
	PDI_min_	PDI_median_	PDI_max_	pTDI_min_	pTDI_median_	pTDI_max_
Entire group	2	30	461	1	15	231
Adults (*n* = 138)	2	13	214	1	6.5	107
Children (*n* = 179)	3	50	461	1.5	25	231

* The level of no concern for nephrotoxicity set by EFSA [4].

**Table 3 toxins-15-00026-t003:** Demographic characteristics of the urine donors in the study groups.

Region Group	Age (Years)	Body Weight (kg)	Urine Donors N and (Gender)
Bavaria			
Children	2–14 (Mean 6)	11–56 (Mean 20.2)	93 (47 m, 46 f)
Adults	26–61 (Mean 39)	46–107 (Mean 75.1)	76 (37 m, 39 f)
Berlin			
Children	2.0–12 (Mean 7.2)	11–40 (Mean 25.7)	27 (12 m, 15 f)
Adults	24–52 (Mean 40.1)	46–135 (Mean 70.2)	66 (23 m, 43 f)
NRW			
Children (Spot urine)	2.4–6.5 (Mean 4.9)	15–33 (Mean 19.2)	50 (25 m, 25 f)
Children (24 h urine)	4.0–6.5 (Mean 4.9)	16–21 (Mean 18.7)	10 (5 m, 5 f)

## Data Availability

Data sharing not applicable.

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
