# Peer review of "Citrinin Exposure in Germany: Urine Biomarker Analysis in Children and Adults"

_toxins, 2022, doi:10.3390/toxins15010026_

Round 1

Reviewer 1 Report

In this article, the authors investigated CIT exposure in young and adult residents of three federal states in Germany. This study is of interest to the researchers and worth to be published. However, there are several points need to be revised or clarified. My detailed comments are as follows:

1.The article only reported the concentrations of CITtotal (sum of CIT and DH-CIT), however, the separate concentrations of CIT and DH-CIT should also be provided.

2.The levels of CITtotal in morning urines and 24h urines were compared, however, what about the dilution of samples? The creatinine-adjusted concentrations might be more useful.

3.Table 2, “pTDI” was ambiguous, “PDImin” and “pTDImin” for Children were incorrect. The “PDImedian” for Children was 214 ng/kg b.w., while “In children, the median exposure of 0.05 μg/kg b.w.”? Line 129.

4. Figure 3 and figure 2 was repetitive, as DI was calculated based on CITtotal levels.

5.The article compared the concentrations of CITtotal in different age groups, what about other demographic factor, such as gender, BMI…

6. The results of this study were compared to previous research, and a table could help the readers easily understand.

Author Response

see attached cover letter

Reviewer 2 Report

This manuscript “Citrinin Exposure in Germany: Urine Biomarker Analysis in Children and Adults” investigates citrinin exposure in German young and adult people. The results indicate non-negligible exposure to citrinin in Germany, mainly in children.

The authors measured citrinin and its metabolite dihydrocitrinone (DH-CIT), It is mentioned that the sum of parent mycotoxin and its metabolite (Ctotal) in urine best reflects exposure to citrinin. But it is not clear whether it is true or not. Each data should be clearly shown, and they should be discussed.

Although the authors mentioned “follow-up biomonitoring studies and investigations aimed to identify the major sources of CIT exposure in food commodities are required.”, I think it better to add some food data in Germany.

Author Response

see attached Cover Letter

Reviewer 3 Report

The paper presents study of the citrinin biomarkers in human urine. The research is correct but in my opinion, there is a lot of valuable data here that have not been used. The sampling was well planned and the analytical protocol was validated so the determinations are correct. However, for the exposure analysis, there is an assumption that each sample came from a model donor. I totally understand that in a relatively simple study like this one you can not introduce all the factors but for sure you can introduce some. The one factor that would be easy enough to take into account and at the same time changing a lot would be standarization of urine (by creatinine or osmolarity). Then the data could be adjusted to more "true" urine production. It would be extremely interesting to see the difference in the exposure assessment using 24h and morning urine from same individuals after these adjustments. Moreover, a lot more could be gained from the study depending on what information was gathered during the sampling. First of all, the authors mention the sampling was within families. Could you try to do more advanced statistics to see whether clusters appear (e.g. related to families). Another interesting parameter to organize statistics would be the diet. I know such data may not be available but if they are, it is better to use it now than save for another paper.

There is something terribly wrong with table 2. Minimum values are higher than medians, and the data is different from the text.

Author Response

See attached Cover Letter

Round 2

Reviewer 2 Report

The revised one should be fine.

Reviewer 3 Report

I recommend accepting the paper after corrections/ explantaions by the authors.